# Genetic and Molecular Basis of Heterogeneous NK Cell Responses against Acute Leukemia

**DOI:** 10.3390/cancers12071927

**Published:** 2020-07-16

**Authors:** Dhon Roméo Makanga, Francesca Da Rin de Lorenzo, Gaëlle David, Catherine Willem, Léa Dubreuil, Nolwenn Legrand, Thierry Guillaume, Pierre Peterlin, Amandine Lebourgeois, Marie Christine Béné, Alice Garnier, Patrice Chevallier, Ketevan Gendzekhadze, Anne Cesbron, Katia Gagne, Béatrice Clemenceau, Christelle Retière

**Affiliations:** 1Etablissement Français du Sang, 44011 Nantes, France; Dhon.Makanga@efs.sante.fr (D.R.M.); francedarin@gmail.com (F.D.R.d.L.); Gaelle.David@efs.sante.fr (G.D.); catherine.willem@efs.sante.fr (C.W.); lea.dubreuil@efs.sante.fr (L.D.); nolwenn.legrand@efs.sante.fr (N.L.); Anne.Cesbron@efs.sante.fr (A.C.); Katia.Gagne@efs.sante.fr (K.G.); 2Université de Nantes, INSERM U1232 CNRS, CRCINA, F-44000 Nantes, France; thierry.guillaume@chu-nantes.fr (T.G.); MarieChristine.BENE@chu-nantes.fr (M.C.B.); patrice.chevallier@chu-nantes.fr (P.C.); Beatrice.Clemenceau@univ-nantes.fr (B.C.); 3LabEx IGO “Immunotherapy, Graft, Oncology”, F-44000 Nantes, France; 4Hematology Clinic, CHU, 44000 Nantes, France; Pierre.PETERLIN@chu-nantes.fr (P.P.); amandinelebourgeois@wanadoo.fr (A.L.); alice.garnier@chu-nantes.fr (A.G.); 5Hematology Biology, CHU, 44000 Nantes, France; 6HLA Laboratory, Department of Hematology and HCT, City of Hope, Medical Center, Duarte, CA 91010, USA; kgendzek@coh.org; 7LabEx Transplantex, Université de Strasbourg, 67000 Strasbourg, France

**Keywords:** KIR, HLA, natural killer cells, repertoire, acute leukemia, CMV

## Abstract

Natural killer (NK) cells are key cytotoxic effectors against malignant cells. Polygenic and polymorphic Killer cell Immunoglobulin-like Receptor (KIR) and HLA genes participate in the structural and functional formation of the NK cell repertoire. In this study, we extensively investigated the anti-leukemic potential of NK cell subsets, taking into account these genetic parameters and cytomegalovirus (CMV) status. Hierarchical clustering analysis of NK cell subsets based on NKG2A, KIR, CD57 and NKG2C markers from 68 blood donors identified donor clusters characterized by a specific phenotypic NK cell repertoire linked to a particular immunogenetic KIR and HLA profile and CMV status. On the functional side, acute lymphoblastic leukemia (ALL) was better recognized by NK cells than acute myeloid leukemia (AML). However, a broad inter-individual disparity of NK cell responses exists against the same leukemic target, highlighting bad and good NK responders. The most effective NK cell subsets against different ALLs expressed NKG2A and represented the most frequent subset in the NK cell repertoire. In contrast, minority CD57^+^ or/and KIR^+^ NK cell subsets were more efficient against AML. Overall, our data may help to optimize the selection of hematopoietic stem cell donors on the basis of immunogenetic KIR/HLA for ALL patients and identify the best NK cell candidates in immunotherapy for AML.

## 1. Introduction

Natural killer (NK) cells are key cytotoxic effectors of innate immunity against malignant [1] and virally infected cells [2]. They are able to sense the absence or the low expression of HLA class I molecules on cancer or virally infected cells via Killer cell Immunoglobulin-like Receptor (KIR) [3]. On the basis of their recognition of HLA class I, a strong interest in NK cells is focused on hematopoietic stem cell transplantation (HSCT) to optimize the graft-versus-leukemia (GvL) effect. In the context of acute leukemias, NK cell-mediated immunosurveillance requires the establishment of an immunological synapse between the NK cell and the leukemia target, involving a large number of interactions between inhibitory and activating NK receptors and their ligands expressed by leukemic cells. Inhibitory receptors mainly include KIR and NKG2A receptors, which are specific to HLA class I molecules. KIRs are clonally expressed on the surface of mature NK cells. In addition, inhibitory KIRs and autologous HLA class I interactions contribute to the functional education of NK cells [4]. Thus, only KIR^+^ NK cells that have encountered their ligand during development will be educated to recognize their absence [5]. The inhibitory NKG2A receptor is expressed on the surface of immature NK cells and ensures NK cell-mediated immunosurveillance through interaction with the HLA-E molecule [6] before KIR expression. Moreover, NK cells express a wide range of activating receptors that recognize induced or upregulated molecules on tumor cells. Activating NK receptors known to be involved in NK cell activation include NKG2D, DNAM-1, 2B4 and NCRs such as NKp30, NKp44 and NKp46. NKG2D recognizes MICA/B and ULBP ligands, DNAM-1 recognizes PVR and Nectin-2 ligands, and 2B4 recognizes CD48 [7]. NCRs recognize different ligands on the cell membrane that can be induced in response to stress or pathogen infection [8,9,10,11] (Table 1). Cognate ligands are heterogeneously expressed on leukemias. It has been reported that the absence of expression or downregulation of HLA class I molecules, as well as the presence of activating ligands on the surface of leukemia cells, were associated with the reduction of relapse incidence and improved overall survival of patients with leukemia [12,13].

We previously documented the beneficial role of KIR and HLA incompatibilities between the donor and the recipient, which significantly limited relapse in the context of haploidentical HSCT [14]. We showed that KIR and HLA incompatibilities participate in the activation and earlier differentiation of NK cells associated with more graft-versus-host-disease (GvHD) and less relapse [14]. However, we are not yet able to define the most effective NK cell populations against leukemia, depending on the nature of acute lymphoblastic or myeloid leukemia.

KIR and HLA genes participate in the structural and functional formation of the NK cell repertoire. The KIR and HLA gene families constitute the most polymorphic and polygenic receptor-ligand pair in humans, and their functional interactions drive tremendous NK cell diversity with a limited number of germline-encoded genes [15]. According to the NK cell differentiation model described by Bjorsktrom et al. [16], NK cell subsets can be defined on the basis of early differentiation markers, such as NKG2A and KIR, and the terminal differentiation CD57 marker [17]. Indeed, during their development, immature CD56^bright^ NK cells express NKG2A and then gradually lose the intensity of expression of CD56, resulting in a pool of CD56^dim^ NK cells [18,19]. At this stage, NK cells can lose the expression of NKG2A and/or acquire KIR and/or CD57 markers in a non-coordinated manner, leading to mature CD56^dim^ NK cell subsets. Human cytomegalovirus (CMV) drastically changes the NK cell repertoire, favoring a massive and lifelong expansion of adaptive-like NK cells that express activating CD94/NKG2C receptors; these cells constitute the most mature subset [20,21,22]. These amplified NKG2C^+^ NK cells preferentially co-express inhibitory KIR, such as KIR2DL2/3 or KIR2DL1, which is involved in functional NK cell education [23]. Of note, peptide-specific recognition of CMV strains controls adaptive NK cells, which, in part, explains why all CMV seropositive individuals do not display this expanded memory-like NK subset [24].

In this study, we hypothesized that KIR and HLA immunogenetic markers and CMV status drive the NK cell subset distribution and that NK cell subsets do not share similar degranulation potential against leukemias, depending on the nature thereof. Thus, we carried out an in-depth investigation of the anti-leukemic potential of NK cell subsets against a panel of acute myeloid and lymphoid leukemia cell lines and primary leukemic cells, taking into account KIR and HLA genetic parameters, NK cell development stages and the CMV status of healthy blood donors.

## 2. Results

### 2.1. Lymphoid Cell Lines More Strongly Stimulated NK Cell Degranulation than Myeloid Cell Lines

To explore the diversity of the anti-leukemia potential of NK cells, we initially investigated NK cell degranulation from a validation cohort of healthy blood donors (n = 14) against a panel of myeloid and lymphoid cell lines (Figure 1A). Lymphoid cell lines, including Burkitt cell lines and acute lymphoblastic leukemia (ALL), more strongly triggered NK cell degranulation than myeloid cell lines, which mainly included acute myeloid leukemia (AML) cell lines (Figure 1A). Given the diversity of NK cell responses based on the target cell nature, we aimed to characterize the phenotype of all target cell lines for the expression of NK receptor ligands known to be involved in the modulation of NK cell functions. Thus, we determined the expression of HLA class I and HLA-E, which are ligands of inhibitor NK receptors, and ULBP1-6, MICA/B, Nectin-2 (CD112), PVR (CD155), CD48 and B7-H6, which are ligands of activating NK receptors. Hierarchical clustering analysis of ligand expression was performed with the Genesis^®^ software, taking into account the mean fluorescence intensity (MFI) of each NK receptor ligand (Figure 1B). Two clusters of target cells can be identified according to trends based on lymphoid and myeloid origins. The first cluster, which mainly includes lymphoid cell lines (except for KG1), showed higher expression of HLA class I, HLA-E, ULBP1/2/5/6 and CD48 ligands than myeloid cell lines. The second cluster, which mainly includes myeloid cell lines (excluding HP Ball), showed higher expression of ULBP3/4, MICA/B, Nectin-2 (CD112) and PVR (CD155) ligands than lymphoid cells (Figure 1B). This dichotomy of ligand expression between myeloid and lymphoid cells leads to different receptor-ligand interactions between NK cells and their cellular targets.

To undertake our study on a broader cohort of healthy donors, we focused our investigation on H9, an acute lymphoblastic leukemia (ALL) cell line, and KG1, an acute myeloid leukemia (AML) cell line, as two representative cell lines that are differently recognized by NK cells.

According to the immunophenotypic analysis, the H9 ALL cell line expressed more ULBP2/5/6, CD48 and B7-H6 than the KG1 AML cell line, which, in contrast, expressed Nectin-2 (CD112) and MICA/B, which were slightly expressed and not expressed on the H9 cell line, respectively (Appendix A). Both H9 and KG1 target cells similarly expressed activating ULBP1 and PVR (CD155) ligands (Appendix A). The expression (MFI) of activating NKG2D (Figure 1C), DNAM-1 (Figure 1D) and 2B4 (Figure 1E) were significantly more downregulated on H9 than those on KG1-stimulated NK cells compared with steady-state. These results suggest that activating NKG2D, DNAM-1 and 2B4 receptors were potentially more engaged with their cognate ligands expressed on H9 ALL.

### 2.2. Inter-Individual Diversity of NK Cell Responses against Leukemia Target Cells

We further addressed the inter-individual diversity of NK cell responses from a broader cohort of 68 healthy blood donors. We showed that the H9 ALL cell line stimulated NK cell degranulation to a significantly greater extent than the KG1 AML cell line, highlighting bad and good responders against both target cell lines (Figure 1F). In addition, the best responders against H9 were not systematically good responders against KG1 (Figure 1G), and inversely, good responders against KG1 were not all good responders against H9 (Figure 1H). Our results show a broad inter-individual diversity of NK cell responses against leukemic cells. Moreover, NK cell responsiveness was dependent on the nature of leukemic cells, suggesting that different NK cell subsets are engaged depending on the nature of leukemic cells. These observations led us to explore the intra-individual diversity of NK cell responses.

### 2.3. The Significant Impact of Immunogenetic HLA/KIR Polymorphisms and CMV Status on the Phenotypic Structuration of the NK Cell Repertoire

To further investigate the intra-individual diversity of NK cell responses, we refined our investigations to take into account immunogenetic KIR/HLA polymorphisms and CMV status. 

According to the NK cell differentiation model described by Bjorkstrom et al. [16], we defined NK cell subsets on the basis of NK cell maturation stages [18] by targeting NKG2A, KIR2DL2/3 (denoted hereon as KIR), NKG2C and CD57 markers (Appendix A), and we determined the frequency of each NK cell subset by multiparameter flow cytometry (MFC) from 68 healthy blood donors, who were divided according to KIR and HLA genetics and CMV status (Appendix A). We found that NKG2A^+^ KIR^−^ CD57^−^ and NKG2A^−^ KIR^−^ CD57^−^ NK cells were the two main subsets in terms of frequency, while NK cell subsets expressing KIR and CD57 markers were represented in low proportions (Appendix A).

Hierarchical clustering analysis of the NK cell repertoire by Genesis^®^ software led us to identify nine clusters of individuals based on NK cell frequencies (Figure 2A). Each cluster was characterized by a specific NK cell repertoire (Figure 2B) linked to a particular genetic KIR and HLA profile and CMV status (Figure 2C). Thus, B^+^ KIR genotypes favored a higher frequency of NKG2A^+^ KIR^+^ NK cell subsets compared with AA KIR genotypes; the A3/A11 environment favored higher frequencies of mature CD57^+^ and CD57^+^KIR^+^ NK cells; C2C2 and Bw4 environments favored a higher frequency NKG2A^+^ KIR^−^ CD57^−^ NK cells, and a higher frequency of NKG2A^+^ CD57^+^KIR^−^ NK cells was favored in the Bw4 environment (Figure 2D). In accordance with the literature, the NKG2C^+^ NK cell frequency was significantly higher in CMV^+^ individuals (Figure 2D). Overall, we observed that each NK cell subset was favored by a specific KIR genotype and/or HLA class I environment or by the presence of CMV.

### 2.4. NKG2A^+^ NK Cell Subsets are the Most Efficient against ALL Target Cells

We carried out an in-depth investigation of the degranulation of NK cell subsets from 68 healthy blood donors against the H9 ALL cell line. Because all CMV^+^ individuals (n = 32) from our cohort of healthy blood donors did not necessarily exhibit the amplification of NKG2C^+^ NK cells, we excluded the NKG2C marker in this part of the study to define eight NK cell subsets on the basis of NKG2A, KIR and CD57 expression. Hierarchical clustering analysis (Genesis^®^) of the degranulation of eight NK cell subsets stimulated with the H9 cell line defined six clusters of individuals (Figure 3A). The superposition of all clusters depending on the immunogenetic KIR and HLA environment and CMV status highlighted specific profiles (Figure 3B). We observed a disparity of whole NK cell degranulation of different clusters, outlining good (C1, C3 and C4 clusters) and bad responders (C2, C5 and C6 clusters), with a significant difference between C1 and C5 (66.40% versus 42.33%, *p* < 0.01) and C4 and C5 clusters (56.82% versus 42.33%, *p* < 0.05) (Figure 3C). Overall, we observed that CMV^−^ KIR2DS1^−^ individuals were the best responders against the H9 ALL cell line (59.35% versus 50.12%, *p* = 0.0047) (Figure 3D). The shared attribute among the most effective NK cell subset against the H9 cell was NKG2A expression (Figure 3E). Interestingly, the degranulation of each NK cell subset in each individual cluster showed that the NKG2A^+^ KIR^−^ CD57^−^ NK cell subset was efficient in all clusters (Figure 3F). Of note, NKG2A^+^ KIR^−^ CD57^−^ NK cell frequency significantly correlated with the degranulation of whole NK cells (*p* < 0.0001) (Figure 3G). This result is in accordance with the prevalence of the NKG2A^+^ KIR^−^ CD57^−^ NK cell subset in the NK cell repertoire.

We further investigated the degranulation potential of NK cell subsets against COE-B primary ALL blasts isolated from a patient. Hierarchical clustering analysis of the degranulation of NK cell subsets from 51 healthy blood donors determined six clusters of individuals (Appendix A). We observed a disparity of whole NK cell degranulation of different clusters (Appendix A). The best NK responders were C2C2 individuals (24.90% C1C1, 23.11% C1C2 and 34.06% C2C2 individuals; *p* = 0.014 between C1C1 and C2C2 individuals, *p* = 0.004 between C1C2 and C2C2 individuals) (Appendix A). As observed with the H9 ALL cell line, the most effective NK cell subsets against primary ALL blasts expressed NKG2A (Appendix A). NKG2A^+^ KIR^−^ CD57^−^ NK cell frequency significantly correlated with the degranulation of the whole NK cells (*p* < 0.0001) (Appendix A). Overall, the best NK responders present a C2C2 environment that favors a high frequency of NKG2A^+^ KIR^−^ CD57^−^ NK cells, the most efficient NK cells against ALL blasts.

### 2.5. KIR^+^ and CD57^+^ NK Cell Subsets are the Most Efficient against AML Target Cells

In parallel to our investigations on NK cell responses against ALL targets, we carried out an in-depth evaluation of these responses against the AML KG1 cell line and two primary AML blasts from the cohort of 68 healthy blood donors. After KG1 stimulation, we determined eight individual clusters based on the degranulation of the eight studied NK cell subsets using hierarchical clustering analysis (data not shown). C1, C2, C3 and C5 individual clusters presented a strong whole NK cell degranulation in contrast to C4, C6, C7 and C8 individual clusters (*p* < 0.0001) (Figure 4A). Interestingly, good responders (C1, C2, C3 and C5) were predominantly C1C2 (*p* = 0.0021) in contrast to bad responders (C4, C6, C7 and C8) (Figure 4B). We did not observe an impact of the KIR (AA versus B+) genotype, HLA-A3/A11 and HLA-Bw4 environments or CMV status on whole NK cell degranulation against the KG1 AML cell line (Figure 4C). However, in accordance with the profile of the best NK responders, C1C2 individuals were better responders against KG1 compared with C1C1 individuals (30.74% versus 20.47%, *p* = 0.0038) (Figure 4C). Moreover, among C1C2 individuals, AA KIR genotyped individuals were significantly better responders than B+ KIR genotyped individuals (36.08% versus 26.47%, *p* = 0.0082) (Figure 4C). The most effective NK cell subsets against the KG1 AML cell line shared the CD57 marker, as observed in good (whole NK cell degranulation > 20%) and bad responders (whole NK cell degranulation < 20%) and all studied donors (Figure 4D). In focusing on the frequency of studied NK cell subsets, a significant difference was observed between good and bad responders for the KIR^+^ NKG2A^−^ CD57^−^ NK cell subset (7.45% versus 5.29%, *p* = 0.0139) and KIR^+^ NKG2A^+^ CD57^−^ (4.99% versus 3.29, *p* = 0.0198) (Figure 4E). These results suggest a functional advantage of KIR^+^ NK cells against the KG1 cell line, even though they did not constitute the best effectors.

Moreover, hierarchical clustering analysis of the degranulation of the eight NK cell subsets against two AML blasts (WAL-C and JAI-S) led us to identify seven clusters of individuals, as illustrated for the most representative AML blasts (WAL-C) in Figure 5A. The comparison of whole NK cell degranulation against primary AML blasts among these seven clusters revealed bad (C2, C5, C6, C7) and good (C1, C3, C4) responders (Figure 5B). The analysis of the immunogenic KIR/HLA profile of bad and good responders indicated that the best responders were significantly more likely to be A3/A11^+^C2^+^ CMV^−^ than their bad counterparts (*p* = 0.024) (Figure 5C). These results were confirmed for all individuals (6.62% versus 3.92%, *p* = 0.033) (Figure 5D). The most effective NK cell subsets against primary AML cells expressed KIR (Figure 5E). NKG2A^+^ CD57^+/−^ KIR^+^ cell subsets constituted the main effective NK cells against primary AML targets, in accordance with results observed against the AML KG1 cell line. Efficient NK cell subsets against AML targets were not well represented in the NK cell repertoire, explaining the low degranulation of whole NK cells observed against AML. In contrast, in terms of frequency, the most predominant NKG2A^+^ KIR^−^ CD57^−^ and NKG2A^−^ KIR^−^ CD57^−^ NK cell subsets harbored very low degranulation against AML cells (Figure 5E).

### 2.6. Hyporesponsiveness of CMV-Driven NKG2C^+^ NK Cell Subsets against Leukemia Target Cells

For the study of NG2C^+^ NK cell functionality, we focused only on CMV^+^ individuals with the amplification of NKG2C^+^ NK cells (higher than 12.28%, corresponding to the mean value). From the 68 healthy blood donors investigated in this study, 32 individuals were CMV seropositive, and only 8 CMV^+^ individuals presented an amplification of NKG2C^+^ NK cell subsets (Figure 6A). From these eight CMV^+^ individuals, two clusters of individuals were defined by hierarchical clustering analysis of the frequency of four NKG2C^+^ NK cell subsets based on NKG2A, KIR and CD57 marker expression (Figure 6B,C). The first cluster harbored C1^+^ individuals (three C1C1 individuals and one C1C2 individual) with an amplification of the NKG2C^+^ KIR2DL2/3^+^ CD57^+/−^ NK cell subsets (Figure 6C). In contrast, the second cluster harbored C2^+^ individuals (three C2C2 individuals and one C1C2 individual) with an amplification of the NKG2C^+^ KIR2DL2/3^−^ CD57^+/−^ NK cell subsets (Figure 6C), in accordance with the preferential amplification of educated NK cells via inhibitory KIR [23].

The degranulation of NKG2C^+^ KIR2DL2/3^−^ CD57^−^ NK cell subsets against ALL cells (H9 cell line and primary ALL blasts) was significantly higher than that of NKG2C^+^ KIR2DL2/3^+^ CD57^+^ NK cell subsets (64.46% versus 30.75%, *p* = 0.0084, against H9, and 56.53% versus 31.87%, *p* = 0.0011, for COE-B blasts), regardless of the HLA-C environment of NK cells (C1^+^ or C2^+^) (Figure 6D). Of note, the C2^+^ NK responses of all NK cell subsets against ALL blasts were higher than the C1^+^ counterpart, in accordance with our previous results obtained from the whole cohort. In contrast, the degranulation of all NKG2C^+^ NK subsets was low against the AML KG1 cell line and blasts. Overall, we observed that NKG2C^+^ KIR2DL2/3^−^ CD57^−^ NK cell subsets (and probably KIR2DL1^+^), amplified in C2^+^ CMV^+^ individuals, were the most effective subsets against ALL H9 and primary cells, as well as against primary AML targets.

Several research groups consider memory-like NKG2C^+^ NK cells to be attractive anti-leukemia candidates for immunotherapy [25]. Thus, we further examined the degranulation potential of memory-like NKG2C^+^ NK cells in these eight CMV^+^ individuals. We compared that to the degranulation of eight NK cell subsets defined on the basis of NKG2A, KIR and CD57 markers. The degranulation of NKG2C^+^ NK cells was low against H9, KG1 and primary AML blasts (Figure 6E). Although the whole NKG2C^+^ NK cell population exhibited better responses against primary ALL blasts (Figure 6E), this population seems to be not as attractive for harnessing anti-leukemia immunity.

## 3. Discussion

On the basis of clinical data, NK cells appear to play a crucial role in the eradication of acute leukemia [26,27,28]. However, the ability of NK cells to mediate a response against tumor cells such as leukemia cells depends on the presence of ligands that are recognized by NK receptors on tumor cells [12,13,26,29]. In this study, a broad diversity of NK cell responses was observed in vitro against myeloid and lymphoid cell lines. Most of the lymphoid cell lines triggered stronger NK cell responses than myeloid cell lines. Furthermore, a broad inter-individual disparity of NK cell responses was observed against the same leukemic target, highlighting bad and good NK responders. In addition, the best responders against one target were not systematically good responders against another. These results showed that the diversity of NK cell responses against leukemic cells were inter- and intra-individual. On the basis of NKG2A, KIR2DL2/3, CD57 and NKG2C markers, we determined different clusters of individuals who shared a common NK cell repertoire linked to the immunogenetic KIR/HLA profile and CMV status. We found that the NKG2A^+^ KIR2DL2/3^−^ CD57^−^ NK cell subset was predominantly represented in the majority of NK cell repertoires and more significantly in C2C2 Bw4^+^ genotyped individuals. Of note, C2C2 individuals are more represented in the African population (up to 40%) than Caucasian (20%) and Asian (less than 5%) populations. HLA-C of the C2 group and HLA-Bw4 ligands contribute to the education and maturation of KIR2DL1 and KIR3DL1 NK cells, respectively [4]. As KIR2DL1 and KIR3DL1 expression were not investigated in this study, it is possible that a proportion of NKG2A^+^ KIR2DL2/3^−^ CD57^−^ NK cells express KIR2DL1 and KIR3DL1 receptors. In contrast, NKG2A^−^ KIR2DL2/3^+^ CD57^−^, NKG2A^−^ KIR2DL2/3^+^ CD57^+^ and NKG2A^−^ KIR2DL2/3^−^ CD57^+^ NK cell subsets were represented in low frequencies in NK cell repertoires. NKG2A^−^ KIR2DL2/3^+^ CD57^−^ NK cells were favored in A3/A11^−^/Bw4^−^ genotyped individuals. No significant difference between C1 and C2 ligands was observed in the frequency of this KIR2DL2/3^+^ NK cell subset. Thus, the absence of A3/A11 and Bw4 ligands suggests that C1 or C2 ligands favored the education and development of the NKG2A^−^ KIR2DL2/3^+^ CD57^−^ NK cell subset since it has been reported that the KIR2DL3 receptor can strongly recognize the C1 ligand and recognizes the C2 ligand with low affinity [30]. NKG2A^−^ KIR2DL2/3^+^ CD57^+^ and NKG2A^−^ KIR2DL2/3^−^ CD57^+^ NK cell subsets were particularly favored in A3/A11^+^ genotyped individuals. As A3/A11 HLA class I molecules, originally identified as ligands for the KIR3DL2 inhibitory receptor [31], participate in the development of KIR3DL2^+^ NK cells, we can speculate that a proportion of NKG2A^−^ CD57^+^ NK cells express KIR3DL2. Our data are placed in a broad body of literature on the structuration of the NK cell repertoire [30,32,33,34,35,36,37], with the notably recent study of Pfefferle et al., showing a well-maintained NK cell repertoire over time [18]. Indeed, the intra-lineage plasticity maintains phenotypic and functional NK cell homeostasis, and the acquired phenotype determines functional potential in NK cells. Because the KIR2DL2/3^+^ NK cell represents the most prevalent NK cell subset in the KIR NK cell repertoire, we focused only on the KIR2DL2/3 inhibitory receptor in this study. However, further investigations are needed to evaluate the role of other inhibitory KIR receptors, such as KIR2DL1, KIR3DL1 and KIR3DL2, in anti-leukemia NK cell responses, particularly against AML.

An in-depth functional analysis identified NKG2A^+^ KIR^−^ CD57^−^ NK cells as the most effective NK cell subsets against the ALL H9 cell line and ALL blasts, as previously reported [38]. The fact that NKG2A^+^ NK cells were represented at a high frequency in the NK cell repertoire explains the strong response of these subsets against ALL targets. Moreover, Forslun et al. reported that NKG2A^+^ KIR^−^ CD57^−^ NK cells exhibit more dynamic migration, which is associated with increased target cell conjugation and a higher probability of killing compared with NKG2A^−^ KIR^−^ CD57^−^ NK cells. The authors suggested that the ability of NKG2A^+^ NK cells to form more conjugates resulted in their increased motility, leading to more encounters with target cells [39]. In our study, ALL targets expressed HLA-E molecules. We can speculate that this inhibitory effect of NKG2A is probably counterbalanced by the strong activating signals mediated through the interaction of NKG2D and DNAM-1 and their cognate ligands expressed on ALL targets. Thus, the cellular mechanism behind the efficient response mediated by these NK cells expressing NKG2A, particularly the response against lymphoid cells, remains unclear. Of note, good responders against ALL targets were either KIR2DS1^−^ or C2C2 individuals, the latter harboring a higher frequency of NKG2A^+^ KIR^−^ CD57^−^ NK cells. These parameters are important, as they can help in choosing donors of HSC that harbor more efficient and abundant NKG2A^+^ KIR^−^ CD57^−^ NK cells to treat acute lymphoblastic leukemia.

In the context of haploidentical HSCT, numerous studies have reported that NK cell-mediated alloreactivity is induced by donor KIR^+^ NK cells [40,41,42]. In a previous study, we documented that KIR/HLA incompatibilities were associated with the more differentiated phenotype of the NK cell repertoire at the time of GvHD, limiting the incidence of relapse after haploidentical HSCT [14]. In line with these previous studies, we report here that more differentiated NK cells expressing KIR and CD57 markers were the most effective NK cells against AML targets, including primary AML blasts. Although KIR^+^ and CD57^+^ NK cells represented the most effective NK cells against AML targets, the low frequency of these NK cell subsets explains why AML targets did not trigger whole NK cell degranulation. Conversely, NKG2A^+^ KIR^−^ CD57^−^ NK cells harbor a low responsiveness against AML targets. This observation is in line with the study of Nguyen et al., who showed that increased frequency of immature NKG2A^+^ NK cells during NK cell repertoire reconstitution after HSCT did not produce a GvL effect in AML patients [43]. In contrast, KIR^+^ NK cells with the CD57 phenotype display increased functional potential associated with the increased transcription of genes involved in adhesion and immune synapse [18]. Altogether, these data suggest that KIR^+^ and CD57^+^ NK cell subsets may be more adapted to interact with and eliminate AML target cells. Among different therapeutic approaches developed to enhance NK cell cytotoxicity against cancer [44], the expansion of allogeneic NK cells constitutes an appealing approach [45]. Thus, the expansion of selected NK cells based on KIR and CD57 expression should constitute promising cell immunotherapy after HSCT to improve the GvL effect in AML patients. Moreover, new insights open avenues in NK cell-based immunotherapy, in particular, chimeric activating receptor (CAR) NK cells [46]. We can imagine genetically engineering the most efficient KIR^+^ and CD57^+^ NK cell subsets against AML to harness their ability to kill these targets.

Furthermore, we documented the massive amplification of adaptive and more differentiated NKG2C^+^ NK cells in seropositive CMV individuals, consistent with previous reports [20,21,22,23]. NKG2C^+^ KIR2DL3^+^ CD57^+/−^ cells were preferentially expanded in C1^+^ individuals, whereas NKG2C^+^ KIR2DL3^−^ CD57^+/−^ NK cells were preferentially expanded in C2^+^ individuals, demonstrating the important role of the HLA-C environment in the expansion of such NK cell subsets, as previously observed [20,23,47,48]. Lopez-Vergès et al., reported that during CMV infection, there is a unique expansion of NKG2C^+^ CD57^+^ NK cells, and they proposed that CD57 might provide a marker of “memory” NK cells in response to CMV infection [49]. Our study identified expanded NKG2C^+^ NK cells with and without CD57 expression, suggesting the presence of phenotypic intra-individual diversity of the NKG2C^+^ NK cell repertoire in CMV seropositive individuals. Functionally, adaptive NKG2C^+^ NK cell responses were more diverse than initially expected [40]. Interestingly, we noted low responsiveness of the well-described adaptive and memory-like NKG2C^+^ KIR^+^ CD57^+^ NK cells against all leukemia targets, in contrast to other NKG2C^+^ NK cell subsets. Our study identified NKG2C^+^ KIR^−^ CD57^−^ NK cells as the most effective NKG2C^+^ NK cell subset. Of note, taking into account all NK cell subsets investigated on the basis of KIR, CD57 and NKG2A expression, the NKG2C^+^ NK cell subset was among the least efficient NK cell subset against all leukemia targets. Merino et al. recently showed that chronically stimulated adaptive NK cells were dysfunctional when challenged with tumor cells [50]. Although this NK subset displays attractive characteristics, we observed limited anti-tumor efficiency.

In conclusion, our data may have evident clinical implications, as they can be used to optimize the selection of HSC donors on the basis of immunogenetic KIR and HLA for ALL patients and identify the best NK cell subsets in immunotherapy to cure AML patients.

## 4. Materials and Methods

### 4.1. Peripheral Blood Mononuclear Cells (PBMCs), Cell Lines and Primary Target Cells

Peripheral blood mononuclear cells (PBMCs) were isolated as previously described [32]. All blood donors were recruited at the Blood Transfusion Center (EFS, Nantes, France), and informed consent was given by all donors. A panel of myeloid and lymphoid cell lines (n = 10) and primary leukemic cells (n = 2) were used to investigate the potential of NK cell degranulation. Cell lines included acute myeloid leukemia (AML) cell lines (n = 4): HL60, KG1, NB-4 and OCI-AML3; T-acute lymphoblastic leukemia (T-ALL) cell lines (n = 4): Jurkat, Molt-4, HP B-ALL and H9; and Burkitt Lymphoma cell lines (n = 2): Raji and Daudi. KG1, NB4 and OCI-AML3 cell lines were generously supplied by Dr Nicolas Dulphy (UMRS-1160 INSERM, Paris, France). Cell lines were cultured in RPMI 1640 medium (Gibco, Paisley, Scotland, UK) containing glutamine (Gibco) and penicillin-streptomycin (Gibco) and supplemented with 10% fetal bovine serum (Gibco). Mycoplasma tests performed by PCR were negative for all cell lines. Blood samples from two patients with acute leukemia were used to isolate leukemic blasts by the same method used for PBMC isolation. All patients gave their informed consent to physicians from the clinical hematology department of the Nantes University Hospital (Pr. Patrice Chevallier). Primary leukemic cells included one acute myeloid leukemia (AML) target (blasts > 90%) and one B-acute lymphoblastic leukemia (B-ALL) target (blasts > 95%) named WAL-C and COE-B, respectively. All leukemic cells (lines and blasts) were HLA typed. A declaration of preparation and conservation of these biocollections (DC-2014-2340) has been provided to French Research Minister and has received approval from the IRB (2015-DC-1).

### 4.2. HLA and KIR Genotyping

HLA class I allele assignment and KIR gene content typing were performed as previously described [14] for all volunteer blood donors (n = 68). KIR genetic typing was performed on all donor samples using a KIR multiplex PCR-SSP method as previously described using specific KIR primers provided by Dr. Ketevan Gendzekhadze (City of Hope, Duarte, CA, USA) [51]. More details are provided in Appendix B.

### 4.3. HCMV Serology

CMV serology was carried out by the technical platform of the Virology Department of Nantes University Hospital (Pr. Berthe-Marie Imbert and Dr. Céline Bressollette) using plasmas from donor biocollections. The CMV serological status of donors was determined by the detection of IgG2 anti-CMV antibodies in plasmas using the LIAISON chemiluminescent immune test (DiaSorin, Saluggia, Italie) according to the manufacturer’s instructions [23]. Our cohort of blood donors comprised 35 seronegative CMV individuals (CMV^−^) and 33 seropositive CMV individuals (CMV^+^).

### 4.4. Phenotypic Analysis of NK Cells, Cell Lines and Primary Leukemic Cells by Flow Cytometry

High-resolution immunophenotyping of NK cells, cell lines and primary leukemic cells was determined by eight-color multiparameter flow cytometry (MFC) using the mouse anti-human mAbs presented in Appendix B.

### 4.5. CD107a Mobilization Assay Detected by Flow Cytometry

NK cells from healthy blood donors were pre-incubated with anti-CD107a-BV421 (H4A3, BD Biosciences, Le Pont de Claix, France). CD107a is a sensitive marker for the identification of NK cell activity [52] and can be used for isolation of tumor-cytolytic cells [53]. NK cell degranulation against myeloid and lymphoid cell lines was assessed after incubation for 5 h alone (negative control) or with different target cells (E/T ratio = 1:1 and 2.5 × 10^6^ cells/well) in bottom 96 well plate with brefeldin A (Sigma, Lezennes, France) at 10 μg/mL for the last 4 h. An initial cell surface staining was used to target NK cell subsets by MFC using the following mouse-Abs combination: CD57-FITC, NKG2C-PE, NKG2D-PerCP-Cy5.5, NKG2A-PC7, KIR2DL2/3-APC, CD56-APC-Cy7 and CD3-BV510. A second cell surface staining was used to analyze NK cell activating receptor expression by MFC using the following mouse-Abs combination: 2B4-FITC, NKp30-PE, DNAM-1-PerCP-Cy5.5, NKp44-PC7 and NKp46-APC. As some primary leukemic cells expressed CD56, the NKp46 marker was used to target NK cells in the degranulation assay by MFC using the following mouse-Abs combination: NKG2A-FITC, NKG2C-PE, KIR2DL2/3-PE-Cy5.5, CD57-PC7, KIR2DL2/3-APC and NKp46-APC. All flow cytometry data were collected using a FACSCanto II (BD Biosciences) and analyzed with Flowjo^™^ 10.6 software (BD Biosciences).

### 4.6. Hierarchical Clustering Analysis of NK Cell Phenotype and Degranulation

The hierarchical clustering of investigated NK cell subsets was performed following a complete linkage using the Genesis software [54]. The software is available at www.genome.tugraz.at.

### 4.7. Statistical Analyses

Comparisons of NK cell frequencies between two different series of individuals were performed by Student’s *t*-test. Comparisons of multiple groups were performed by one-way ANOVA using the GraphPad Prism v6.0 software (Ritme informatique, Paris, France). *p*-values < 0.05 were considered statistically significant.

## 5. Conclusions

In conclusion, this study showed a broad inter-individual disparity of NK cell responses against the same leukemic target, highlighting bad and good NK responders. We observed that the diversity of NK cell responses against leukemic cells was inter- and intra-individual. Indeed, on the basis of NKG2A, KIR2DL2/3, CD57 and NKG2C markers, we identified different clusters of individuals who shared a common NK cell repertoire that was linked to the immunogenetic KIR/HLA profile and CMV status. On the functional side, the most effective NK cell subsets against different ALL targets expressed NKG2A and represented the most frequent subset in the NK cell repertoire. In contrast, minority CD57^+^ or/and KIR^+^ NK cell subsets were more efficient against AML. These results may have evident implications in oncohematology to improve both the selection of hematopoietic stem cell donors for ALL patients and immunotherapies to cure AML patients.

## Figures and Tables

**Figure 1 cancers-12-01927-f001:**
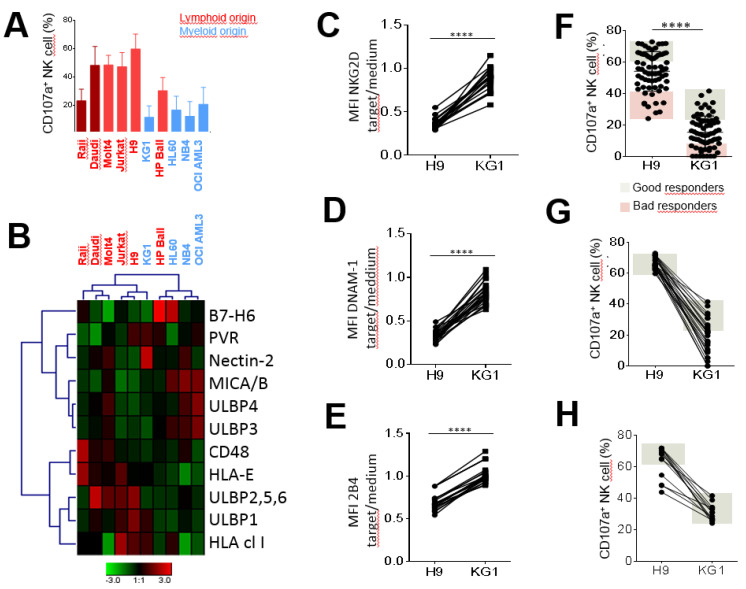
Inter- and intra-individual heterogeneity of NK cell responses against a panel of myeloid and lymphoid target cells. (**A**) Histogram outlining NK (CD3^−^ CD56^+^) cell degranulation observed after 5 h incubation in the presence of myeloid cell lines (blue bar) and lymphoid cell lines (red bar) at an effector/target ratio of 1:1 for 14 representative blood donors. Values are expressed as mean ± SD. (**B**) Heatmap representing the relative MFI of each NK ligand expression on target cell surfaces using Genesis software. Relative MFI represents the ratio of the MFI of each NK ligand on the MFI of isotype control of each mAb. Red and green indicate high expression levels and low expression levels, respectively. The modulation of NKG2D (**C**), DNAM-1 (**D**) and 2B4 (**E**) expression observed after NK cell stimulation in the presence of H9 ALL and KG1 AML cell lines at an effector/target ratio 1:1 from 24 representative healthy blood donors by MFC. (**F**) NK cell degranulation evaluated against H9 ALL and KG1 AML cell lines from a broader cohort of 68 blood donors. Good and bad responders are located in gray and pink zones, respectively. (**G**) NK cell degranulation against H9 and KG1 cell lines is represented for good responders identified against H9. (**H**) NK cell degranulation against H9 and KG1 cell lines is represented for good responders identified against KG1. **** Indicates *p* < 0.0001 (Student’s *t*-test).

**Figure 2 cancers-12-01927-f002:**
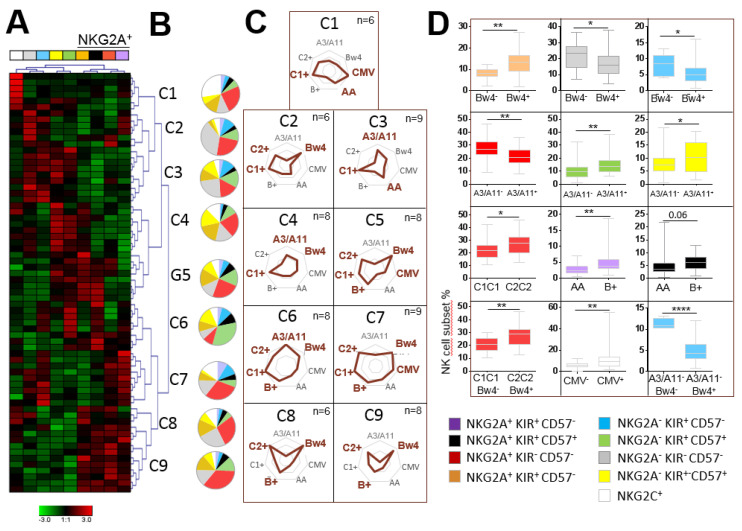
Impact of the immunogenetic KIR/HLA profile and CMV status on the structuration of the NK cell repertoire. (**A**) Heatmap (Genesis^®^) clustering of healthy blood donors (n = 68) from the frequency of nine NK cell subsets targeted by flow cytometry based on NKG2A, KIR, NKG2C and CD57 markers. Each column is dedicated to a defined NK cell subset. Red and green indicate the high and low frequencies of the NK cell subsets, respectively. C1–C9 indicate the nine clusters of individuals. (**B**) Charts representing frequencies of nine investigated NK cell subsets for each cluster. (**C**) Radar charts indicating the number of blood donors for each characteristic (KIR AA or B^+^ genotype, A3/A11, Bw4, C1 and C2 environment and CMV status) per cluster. The KIR/HLA immunogenetic profiles and CMV status impacting each cluster are indicated in brown. (**D**) Whisker graphs of NK cell subset frequencies according to A3/A11, Bw4, C1, and C2 environments, KIR genotype (AA or B^+^) and CMV status investigated in 68 individuals. * Indicates *p* < 0.05, ** indicates *p* < 0.01 and **** indicates *p* < 0.0001.

**Figure 3 cancers-12-01927-f003:**
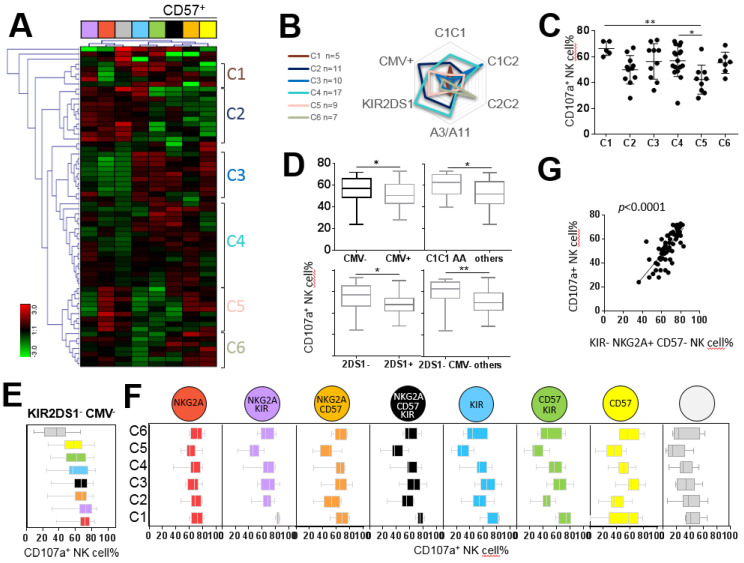
NKG2A^+^ NK cell subsets are the most efficient against the H9 ALL cell line. (**A**) Heatmap (Genesis^®^) clustering of healthy blood donors (n = 68) from the degranulation of eight targeted NK cell subsets. NK cell subset degranulation was assessed after 5 h incubation in the presence of the H9 ALL cell line at an effector/target ratio of 1:1. Each column is dedicated to a defined NK cell subset. Red and green indicate high frequency and low frequency of NK cell degranulation, respectively. C1–C6 indicate the six clusters of individuals. (**B**) Radar charts indicating the number of blood donors for each characteristic (KIR AA or B^+^ genotype, A3/A11, Bw4, C1 and C2 environment and CMV status) per cluster. (**C**) Whisker graphs of whole NK cell degranulation in each cluster. (**D**) Whisker graphs of whole NK cell degranulation according to C1C1 environment, KIR genotype (AA), KIR gene content (2DS1^+^ or 2DS1^−^) and CMV status (CMV^+^ or CMV^−^) investigated in 68 individuals. (**E**) Whisker graphs illustrating the degranulation of the eight investigated NK cell subsets from the best responders, KIR2DS1^−^ CMV^−^, against the H9 ALL cell line. NK cell subsets are classified from the highest to the lowest efficiency. NKG2A^+^ KIR^−^ CD57^−^ NK cells represented the most efficient NK cell subset against the H9 ALL cell line. (**F**) Whisker graphs of NK cell degranulation for each investigated NK cell subset for all clusters. (**G**) Correlation between NKG2A^+^ KIR^−^ CD57^−^ NK cell frequencies and whole NK cell degranulation percentage from 68 individuals. *p*-values are indicated only where a significant *p*-value was obtained (*p* < 0.05). * Indicates *p* < 0.05, ** indicates *p* < 0.01. Spearman’s rank correlation coefficients were calculated, and *p*-values of *p* < 0.0001 were obtained.

**Figure 4 cancers-12-01927-f004:**
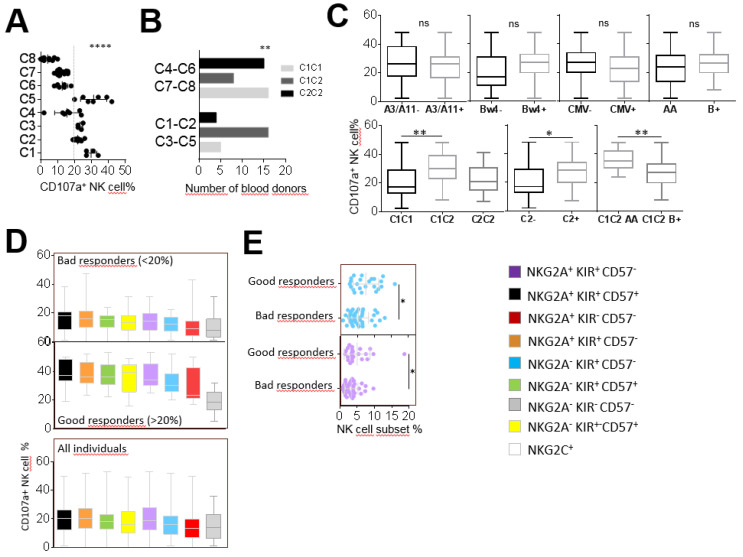
CD57^+^ NK cell subsets are the most efficient against the KG1 AML cell line. (**A**) Dot plots illustrating the whole NK cell degranulation of eight clusters of individuals. (**B**) Histograms indicating the number of individuals according to HLA-C environment (C1C1, C1C2 and C2C2) for good (C1, C2, C3 and C5) and bad (C4, C6, C7 and C8) responders. (**C**) Whisker graphs representing whole NK cell degranulation according to A3/A11, Bw4, C1, C2 environments, KIR genotype (AA or B+) and CMV status investigated in 68 individuals. (**D**) Whisker graphs illustrating the degranulation frequency of eight NK cell subsets against the KG1 AML cell line from bad and good responders and from all individuals (n = 68). NK cell subsets are classified from the highest to the lowest efficiency. **(E)** Dot plots illustrating the NK cell frequency of NKG2A^+^ KIR^−^ CD57^−^ (blue) and NKG2A^+^ KIR^+^ CD57^−^ (purple) NK cell subsets for bad and good responders. *p*-values are indicated only where a significant *p*-value was obtained (*p* < 0.05). * Indicates *p* < 0.05, ** indicates *p* < 0.01, **** indicates *p* < 0.0001.

**Figure 5 cancers-12-01927-f005:**
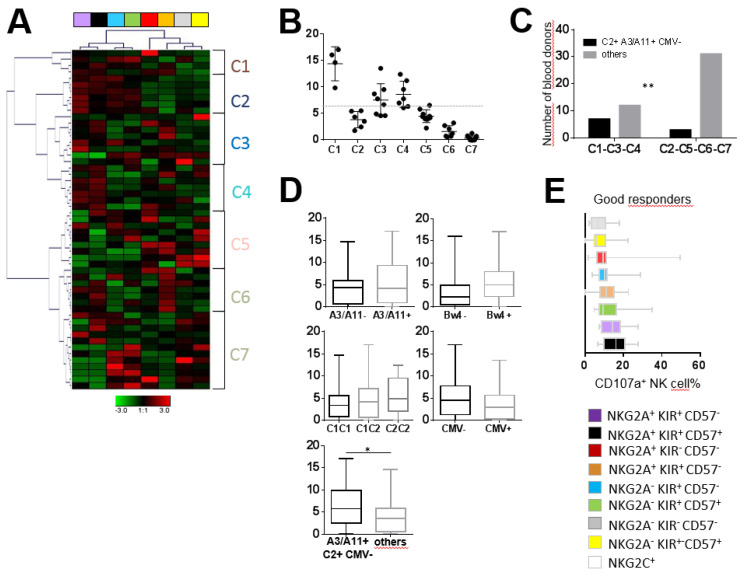
KIR^+^ NK cell subsets are the most efficient against primary AML blasts. (**A**) Heatmap (Genesis^®^) clustering of healthy blood donors (n = 51) from the degranulation of eight targeted NK cell subsets. NK cell subset degranulation was assessed after 5 h incubation in the presence of primary AML target cells at an effector/target ratio of 1:1. Each column is dedicated to a defined NK cell subset. Red and green indicate high frequency and low frequency of NK cell degranulation, respectively. C1–C7 indicate the seven clusters of individuals. (**B**) Dot plots illustrating whole NK cell degranulation of blood donors clustered from C1 to C7. (**C**) Histograms illustrating the number of individuals according to A3/A11, C2 environment and CMV for bad (C2, C5, C6 and C7) and good (C1, C3 and C4) responders. (**D**) Whisker graphs of whole NK cell degranulation according to HLA-Bw4, A3/A11, C1 and C2 environments and CMV status in 51 individuals. (**E**) Whisker graphs of the degranulation frequency of the eight investigated NK cell subsets against primary AML target cells for the good (C1, C3 and C4) responders. NK cell subsets are classified from the highest to the lowest efficiency. *p*-values are indicated only where a significant *p*-value was obtained (*p* < 0.05). * Indicates *p* < 0.05, ** indicates *p* < 0.01.

**Figure 6 cancers-12-01927-f006:**
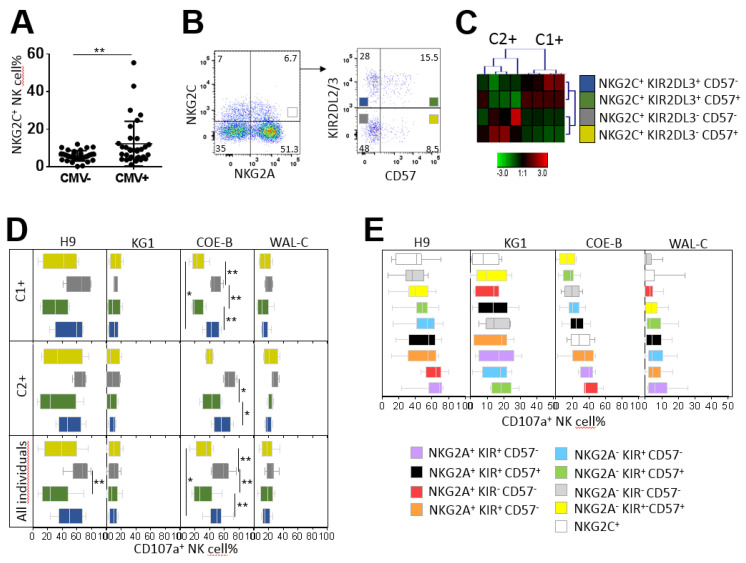
Degranulation potential of CMV-driven NKG2C^+^ NK cell subsets against leukemia target cells. (**A**) Dot plots of NKG2C^+^ NK cell frequencies in CMV^−^ (n = 32) and CMV^+^ (n = 36) individuals. (**B**) Density plots illustrating the cell strategy to target NKG2C+ NK cell subsets expressing or not expressing KIR2DL2/3 and CD57. (**C**) Heatmap (Genesis^®^) clustering of CMV^+^ blood donors with NKG2C NK cell amplification (n = 8) from the frequencies of the four NKG2C^+^ (KIR^+^ CD57^+^, KIR^+^ CD57^−^, KIR^−^ CD57^+^ and KIR^−^ CD57^−^) NK cell subsets. Each column is dedicated to a defined NK cell subset. The color of each square reflects the percentage of the corresponding subset. Red and green indicate high and low frequencies of NK cell subsets, respectively. (**D**) Whisker graphs of the degranulation of the four NKG2C^+^ NK cell subsets against ALL H9 and AML KG1 cell lines, primary ALL COE-B and AML WAL-C blasts for C1^+^ (n = 4), C2^+^ (n = 4) and all individuals (n = 8). (**E**) Whisker graphs of degranulation frequency of the nine investigated NK cell subsets from CMV^+^ individuals (n = 9) with amplification of the NKG2C^+^ NK cell subsets against H9, KG1, CO-E and WAL-C. NK cell subsets are classified from the highest to the lowest efficiency. *p*-values are indicated only where a significant *p*-value was obtained (*p* < 0.05). * Indicates *p* < 0.05 and ** indicates *p* < 0.01.

**Table 1 cancers-12-01927-t001:** Main inhibitory and activating NK cell receptors and their cognate ligands.

NK Receptors	CD Number	Ligands
Activating		
DNAM-1	CD226	Nectin-2 (CD112), PVR(CD155)
2B4	CD244	CD48
NKp30	CD337	B7-H6, BAT3 ^a^, PfEMP1 ^a^
NKp44	CD336	Nidogen-1, PCNA ^a^, HS ^a^, HLA-DP
NKp46	CD335	CFP ^a^, viral HA, HN ^a^ or envelope proteins, PfEMP1 ^a^
NKG2D	CD314	MIC-A ^a^, MIC-B ^a^, ULBP1-6
NKG2C	CD159a	HLA-E
Inhibitory		
NKG2A	CD159a	HLA-E
KIR2DL2/3	CD158b	HLA-C1, HLA-C2 ^b^

^a^ PVR, poliovirus receptor; BAT3, HLA-B-associated transcript 3; PfEMP1, *Plasmodium falciparum* erythrocyte membrane protein 1; PCNA, proliferating cell nuclear antigen; HS, heparan sulfate; CFP, Complement factor P; HA, haemagglutinin; HN, haemagglutinin neuramidase; MIC, major histocompatibility complex (MHC) class I polypeptide-related sequence. ^b^ KIR2DL2/L3 also weakly recognizes HLA-C2 alleles and few HLA-B alleles that bear the HLA-C1 epitope (e.g., HLA-B*4601 and HLA-B*7301).

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
