# Peer review of "Genetic and Molecular Basis of Heterogeneous NK Cell Responses against Acute Leukemia"

_cancers, 2020, doi:10.3390/cancers12071927_

Round 1
Reviewer 1 Report
This is an excellent work demonstrating the involvement of different NK cell subsets in the immune response against acute leukemia. The results may have therapeutic consequence in the field of immunotherapy and/or HSC donor selections. For further improvement of the paper, I suggest following information to be supplemented:
- the figure between the lines 42 and 43 should be positioned elsewhere and title and description of it should be provided as well
- Introduction: please provide a basic overview of NK cell receptors with an appropriate grouping in a form of a table or a figure
- Materials and Methods, 4.5 CD107a mobilization assay: please provide evidence from the literature for the usefulness and comparability of the degranulation assay regarding classical NK cytotoxicity tests. Is the effector/target ratio of 1:1 in this assay the routine?
Author Response
Reviewer 1
This is an excellent work demonstrating the involvement of different NK cell subsets in the immune response against acute leukemia. The results may have therapeutic consequence in the field of immunotherapy and/or HSC donor selections.
Author response: We thank the reviewer for his rewarding comment.
For further improvement of the paper, I suggest following information to be supplemented:
The figure between the lines 42 and 43 should be positioned elsewhere and title and description of it should be provided as well.
Author response: We thank the reviewer for his comment. The figure in question is the graphical abstract. Thus, we removed it from the manuscript and it will appear alongside with the text abstract in the Table of Contents.
Introduction: please provide a basic overview of NK cell receptors with an appropriate grouping in a form of a table or a figure.
Author response: We agree with the reviewer’s comment. Thus, we added different references line 67 to actualize the literature, notably for NKp44 ligands recently identified. Moreover, we added line 74 a table including the main activating and inhibitory NK cell receptors and their ligands.
Materials and Methods, 4.5 CD107a mobilization assay: please provide evidence from the literature for the usefulness and comparability of the degranulation assay regarding classical NK cytotoxicity tests. Is the effector/target ratio of 1:1 in this assay the routine?
Author response: We thank the reviewer to give us the opportunity to reply on this point. The publication of Alter et al. in the Journal of Immunological Methods (2004) was particularly important to show that CD107a is a sensitive marker for the identification of NK cell activity. One year later, Penack et al. in Leukemia (2005) demonstrated that the CD107a mobilization assay can equally well be utilized for isolation of tumor-cytolytic cells. Before to start this study, we assessed 3 E:T ratios (1:1, 2:1 and 3:1) with 0.25x106 PBMC/well in a bottom 96 well plate for all tested conditions. We observed a better degranulation in 5h assay using 1:1 E: T ratio in accordance with Penack’s study in which the cell number (0.2x106 PBMC/well) was similar to our protocol. In the paper of Alter et al., they did not indicate the number of cells/well that can explain the discrepancy of results. This point is an important parameter that can significantly modify the potential of NK cell degranulation. We propose to add these elements line 506 in the part entitled “CD107a mobilization assay detected by flow cytometry” in Materials and Methods section.
Of note, our manuscript has undergone English language editing by MDPI. The text has been checked for correct use of grammar and common technical terms, and edited to a level suitable for reporting research in a scholarly journal.
Reviewer 2 Report
Thank you for giving me a chance to reviewing the manuscript, “Genetic and molecular basis of heterogeneous NK cell responses againstacute leukemia" by Makanga DR, et al.
The authors use the terms “KIR- NK”. This referee is confusing they are using this term for whether all KIRs (including KIR2DL4)-negative NK cells or KIR2DL2/3-negative NK cells. The definition of “KIR” seems to be necessary for the evaluation on the current study.
Author Response
Reviewer 2
Thank you for giving me a chance to reviewing the manuscript, “Genetic and molecular basis of heterogeneous NK cell responses against acute leukemia" by Makanga DR, et al.
The authors use the terms “KIR- NK”. This referee is confusing they are using this term for whether all KIRs (including KIR2DL4)-negative NK cells or KIR2DL2/3-negative NK cells. The definition of “KIR” seems to be necessary for the evaluation on the current study.
Author response: We agree with the reviewer’s comment. We added line 163 that “KIR2DL2/3 is denoted hereon as KIR” to avoid confusing.
English language and style are fine/minor spell check required.
Author response: We thank the reviewer for his useful comment. Our manuscript has undergone English language editing by MDPI. The text has been checked for correct use of grammar and common technical terms, and edited to a level suitable for reporting research in a scholarly journal.
Moreover, to improve the introduction, we added different references line 67 to actualize the literature, notably for NKp44 ligands recently identified. Moreover, we added line 74 a table including the main activating and inhibitory NK cell receptors and their ligands.
Round 2
Reviewer 2 Report
This referee is thinking this manuscript should be acceptable now.